# COVID-19-Related Myocarditis: Are We There Yet? A Case Report of COVID-19-Related Fulminant Myocarditis

**DOI:** 10.3390/biomedicines11082101

**Published:** 2023-07-26

**Authors:** Alessandro Pierri, Giulia Gagno, Alessandra Fluca, Davide Radaelli, Diana Bonuccelli, Laura Giusti, Michela Bulfoni, Antonio P. Beltrami, Aneta Aleksova, Stefano D’Errico

**Affiliations:** 1Cardiothoracovascular Department, Azienda Sanitaria Universitaria Giuliano Isontina, 34139 Trieste, Italy; pierrialessandro8@gmail.com (A.P.); gagnogiulia@gmail.com (G.G.); alessandralucia.fluca@units.it (A.F.); or aaleksova@gmail.com (A.A.); 2Department of Medical, Surgical and Health Sciences, University of Trieste, 34139 Trieste, Italy; davide_radaelli@hotmail.it; 3Department of Legal Medicine, Azienda USL Toscana Nordovest, 55100 Lucca, Italy; diana.bonuccelli@uslnordovest.toscana.it; 4Department of Human Pathology, San Luca Hospital, Azienda USL Toscana Nordovest, 55100 Lucca, Italy; laura.giusti@uslnordovest.toscana.it; 5Institute of Clinical Pathology, Academic Hospital “Santa Maria della Misericordia”, ASUFC, Department of Medicine (DAME), University of Udine, 33100 Udine, Italy; michela.bulfoni@asufc.sanita.fvg.it (M.B.); antonio.beltrami@icloud.com (A.P.B.)

**Keywords:** COVID-19, myocarditis, post-mortem examination, sudden death

## Abstract

There is increasing evidence of cardiac involvement in COVID-19 cases, with a broad range of clinical manifestations spanning from acute life-threatening conditions such as ventricular dysrhythmias, myocarditis, acute myocardial ischemia and pulmonary thromboembolism to long-term cardiovascular sequelae. In particular, acute myocarditis represents an uncommon but frightening complication of SARS-CoV-2 infection. Even if many reports of SARS CoV-2 myocarditis are present in the literature, the majority of them lacks histological confirmation of cardiac injury. Here, we report a case of a young lady, who died suddenly a few days after testing positive for SARS-CoV-2, whose microscopic and genetics features suggested a direct cardiac involvement compatible with fulminant myocarditis.

## 1. Introduction

Coronavirus disease 2019 (COVID-19), caused by the severe acute respiratory syndrome coronavirus-2 (SARS-CoV-2), has been the major health issue worldwide in the past four years. Since the pandemic outbreak, both a higher mortality among patients with previous cardiovascular disease and the potential cardiac involvement related to COVID-19 have been reported [1]. Although severe COVID-19 primarily occurs with fever and very serious respiratory symptoms, it was observed that there was an elevated incidence of myocardial injury, defined as an increase in serum cardiac biomarkers, particularly troponins and natriuretic peptides, above the 99th percentile [1,2]. Notably, the rise of cardiac biomarkers was significantly associated with cardiac dysfunction and malignant arrhythmias, representing an independent predictor of a worse prognosis and fatal outcome [1,2]. The exact mechanisms beyond the elevation of cardiac biomarkers have been long debated. To date, despite intense research, it is still unclear whether SARS-CoV-2 may cause a myocarditis-like disease through cardiac invasion and direct cytotoxicity or if the damage is attributable only to an excessive proinflammatory response as well as to a mismatch between oxygen demand and supply as previously observed in other critically ill patients [1,3,4]. Peretto et al. reported seven cases of patients with a diagnosis of myocarditis and COVID-19 infection; six of them underwent EBM but none of the cases showed a positive SARS-CoV-2 result in myocardial tissue [5]. Apart from the lack of robust pathological evidence of myocarditis due to direct myocardial damage by SARS-CoV-2, the clinical diagnosis of myocarditis in COVID-19 patients is challenging due to overlapping symptoms and signs between myocarditis and other forms of myocardial injury, the lack of electrocardiographic/echocardiographic changes specific for myocarditis and the difficult access to CMR for many patients positive for SARS-CoV-2.

## 2. Case Report

A 30-year-old woman who was not unvaccinated for COVID-19 and without a significant medical history presented to her general practitioner with fever, chills, persistent cough, headache, and sore throat. The physical examination was unremarkable with normal peripheral oxygen saturation at rest (98%) and during a six-minute walk (100%). An empirical treatment with oral amoxicillin–clavulanate and paracetamol was prescribed and a nasopharyngeal swab sample for SARS-CoV-2 testing was taken, which turned out to be positive. Two days later, the patient experienced a sudden onset of profuse sweating and coldness in the extremities and alerted the emergency medical service. Upon arrival of the emergency nurses, the patient was apyretic, had normal oxygen saturation, arterial blood pressure and heart rate. Despite no alarming hints, the emergency nurses proposed to transfer the patient to the Emergency Department (ED) for further care and monitoring, but she refused. The following day, the patient and her husband presented to a local hospital because of clinical deterioration. According to the husband’s report, she was suffering from progressive dyspnoea and consciousness impairment over the past few hours. On arrival at the ED, the patient’s condition suddenly precipitated, and she developed pulseless cardiorespiratory arrest. Advanced cardiopulmonary resuscitation ceased after 20 min of unsuccessful attempts.

Forensic experts performed a complete post-mortem examination in order to establish the precise cause of death. Cervical and thoracic organs were removed en bloc according to Ghon’s technique (Figure 1). The lungs weighted 630 gr (right) and 500 gr (left) and showed oedema and congestion. The presence of segmental and subsegmental pulmonary embolism was excluded. The heart was morphologically normal (it measured cm 11.5 × 9.5 × 3.5 and weighed 300 gr) and showed diffuse subepicardial petechiae in the anterior and posterior wall of the left ventricle (Figure 1). 

The coronaries were normal as well as the valve apparatus. Gross examination of the other organs was unremarkable. Samples of the organs were fixed in 10% buffered formalin and then embedded in paraffin. The histological examination showed bilateral massive pulmonary oedema with mild interstitial lymphocytic infiltration (Figure 2a,b). No features of diffuse alveolar damage, pneumocytes hyperplasia and fibrin thrombi were detected. The heart showed areas of hypercontracted cardiomyocytes with thickening of Z-lines and extremely short sarcomeres as a picture of myofibre breakdown. This breakdown varied from irregular, pathological and eosinophilic cross-bands consisting of segments of hyper-contracted or coagulated sarcomeres to a total disruption of myofibrils; whole cells assumed a granular aspect without visible clear-cut pathological bands. In association with the hypercontracted cardiomyocytes, the heart showed multiple focal areas of mild infiltration of macrophages and lymphocytes as well as erythrocytes (Figure 2c,d). 

Five-micron sections were processed for immunohistochemistry staining; antibodies against SARS-CoV-2 spike protein were tested in heart specimens in which focal lymphocytes infiltration was identified by histological analysis. Infiltrates were further characterized by immunohistochemical staining for CD68, CD45, CD3, CD8 and CD4 performed on an automated staining device. Antigen retrieval techniques and antibody pre-treatment were performed according to the manufacturer’s specifications. Images were acquired using Zeiss AX10 light microscope. Lymphocytic infiltration was characterized by a predominance of CD4+ T lymphocytes (Figure 3a,d). 

In situ hybridization (ISH) was performed using locked nucleic acid (LNA) probes for U6 snRNA (miRCURY LNA Detection probe, Qiagen, Hilden, Germany, cat. No. YD00699002) and SARS-CoV-2 RNA designed to target the sense strand of ORF1ab and spike regions of the viral genome. A scrambled sequence probe (YD00699004) was used as the control. Experiments were performed using a dedicated ISH kit for formalin-fixed paraffin-embedded (FFPE) tissues (Qiagen) according to the manufacturer’s protocol. FFPE tissue slides were deparaffinised in xylene, treated with proteinase-K (15 mg/mL) for 5 min at 37 °C and incubated with either SARS-CoV-2 (40 nM) or U6 probes (2 nM) for one hour at 54 °C in a hybridizer. After washing with SSC buffer, the presence of SARS-CoV-2 RNA was detected using an anti-DIG alkaline phosphatase (AP) antibody (1:500) (Roche Diagnostics, Basel, Switzerland) supplemented with sheep serum (Jackson Immunoresearch, West Grove, PA, USA) and bovine serum albumin (BSA). Hybridization was detected by adding NBT-BCIP substrate (Roche Diagnostics) (Figure 4a–d). 

Finally, samples for real-time polymerase chain reaction (RT-PCR) were extracted from 8 µm sections, which were deparafinized and digested. RNA extraction was performed with Trizol (Invitrogen, Waltham, MA, USA) following the manufacturer’s instructions with some modifications. The quality of total RNA was assessed by measuring the expression of GAPDH. As controls, we used a known positive sample, subjected to the same RNA extraction procedure, and a SARS-CoV-2 MULTITARGET RNA. Detection of SARS-CoV-2 was performed using commercially available primers and probes (Eurofins Genomics, Ebersberg, Germany) for *N* and *ORF1ab* genes and confirmed in heart and lung samples.

## 3. Discussion

Since the first cases of severe COVID-19, a high incidence of myocardial injury has been reported, and it proved to be related with a significant risk of poor outcomes and in-hospital mortality [1,2,3,4,5,6,7]. According to the available literature, systemic hyperinflammatory responses, previous coronary syndrome, pulmonary microembolism, sepsis and aggravated hypoxia represent the best-established pathophysiological mechanisms underlying myocardial injury in patients with severe COVID-19, especially when complicated by acute respiratory distress syndrome (ARDS) [8,9,10]. In contrast, the relationship between SARS-CoV-2 infection and myocardial injury in patients with non-complicated COVID-19 is an area of ongoing debate. In particular, it is not completely understood whether SARS-CoV-2 can directly invade cardiomyocytes thus causing a COVID-19 mediated myocarditis. Since SARS-CoV-2 uses the angiotensin-converting enzyme 2 (ACE2), which is highly expressed on myocardial tissue, to enter cells, the direct damage to cardiomyocytes by the virus has been proposed as a reasonable cause of myocarditis [11]. In addition, recent studies have shown that the human myocardium, from a genetic perspective, meets all the requirements to mediate SARS-CoV-2 cell entry, since a wide range of proteases playing a role in virus–host interaction are expressed at high levels in the heart [12]. Furthermore, several in vitro studies have proved that direct SARS-CoV-2 infection can damage both the contractile and electrical function of cardiomyocytes. The contractile function seems to be affected due to direct damage to myofilaments and to altered expression profiles of genes directly involved in sarcomeric function. Also, SARS-CoV-2 seems to alter the expression of genes involved in energy production, thus adversely affecting myocardial contraction. The electrical dysfunction appears to be linked to an altered homeostasis induced by SARS-CoV-2 infection and results in abnormal generation and propagation of electrical signals [13].

From a clinical perspective, despite several cases of myocardial injuries having been described in the context of COVID-19 disease, most of them have been diagnosed on the basis of imaging findings, especially echocardiography and cardiac magnetic resonance (CMR) [14,15,16]. Despite not providing direct demonstration of COVID-19-induced myocarditis, these diagnostic modalities could offer important clues to this aetiology [17]. For example, in their three-centre experience, Panchal et al. observed that COVID-19-induced myocarditis was characterized by a more diffuse myocardial involvement using T1 and T2 mapping techniques as compared to myocarditis caused by traditional cardiotropic viruses [18]. On the contrary, histopathological evidence of direct myocardial damage is lacking. In particular, Inciardi et al. [14] reported a paradigmatic case of severe myocardial injury in a 53-year-old woman who tested positive for COVID-19. According to CMR diagnostic criteria, namely the Lake Louise criteria, the patient had a diagnosis of acute myocarditis. Nevertheless, the histological confirmation was missing. Interestingly, in those case in which endomyocardial biopsy (EMB) was performed, the presence of the SARS-CoV-2 genome within the myocardium was rarely documented. Sala et al. reported a case of acute myocarditis presenting as a reverse Takotsubo syndrome in a COVID-19 patient undergoing EMB, documenting diffuse T-lymphocytic inflammatory infiltrates with multiple areas of oedema and the absence of the SARS-CoV-2 genome within the myocardium [19]. On the other hand, Tavazzi et al. described a case of cardiogenic shock linked to myocardial localization of SARS-CoV-2, highlighting that the viral particles (detected with an ultrastructural study of the EMB) were found in interstitial cells and not in cardiomyocytes and therefore presuming either a viraemic phase or migration of infected macrophages from the lung [20]. Likewise, Escher et al. assessed the viral presence in EMBs of patients with suspected myocarditis or unexplained heart failure; the SARS-CoV-2 genome was detected in 5 of 104 EMBs, without confirming the infection of cardiac cells [21]. Also, in a cohort study of 39 autopsy cases of patients with COVID-19, in situ hybridization of SARS-CoV-2 RNA demonstrated that the viral genome was more likely localized in interstitial cells or macrophages invading the myocardial tissue rather than in cardiomyocytes. Furthermore, the same study revealed a lack of association between the virus load and the infiltration of mononuclear cells into the myocardium, suggesting that the presence of SARS-CoV-2 in cardiac tissue does not inevitably result in a proinflammatory response consistent with myocarditis [22]. Of note, Bojkova et al. demonstrated that cardiomyocytes are permissive for SARS-CoV-2 infection in cultures, without dwelling on clinical implications. Indeed, their study did not address whether cardiac injury and ventricular dysfunction in COVID-19 patients can be caused by direct infection of cardiomyocytes [23]. To date, only Dolhnikof et al. documented postmortem evidence of viral particles in different cell lineages of the heart, including cardiomyocytes. Specifically, they reported the case of an 11-year-old child with COVID-19-related multisystem inflammatory syndrome who died due to ventricular fibrillation [24]. We presented a case in which a young unvaccinated lady died a few days after testing positive for COVID-19. The clinical history showed the presence of symptoms of heart failure prior to death. The autopsy could not provide any significant findings in terms of the cause of death. The histological slides excluded the presence of significant lung involvement linked to SARS-CoV-2 infection; on the contrary, the heart showed areas of contraction band necrosis associated with lymphocyte and macrophages infiltration compatible with acute myocarditis, which appears to be the most likely cause of death of the patient. This hypothesis was sustained by the autopsy findings of normal epicardial coronary arteries and uninjured valve apparatus components. Indeed, atherosclerotic plaque rupture and destabilization leading to acute coronary syndrome (ACS) and valvular damage, predominantly related to the massive release of circulatory cytokines and chemokines resulting from systemic and uncontrolled inflammation, are well-documented causes of cardiac injury and failure [25]. Furthermore, in situ hybridization and RT-PCR detected the presence of SARS-CoV-2 particles and genomes within the myocardium, suggesting a direct role of SARS-CoV-2 as the aetiological agent responsible for the acute myocarditis in our patient. 

## 4. Conclusions

Although cases of COVID-19 myocarditis have been widely described in the recent literature, there are limited data proving an active role of SARS-CoV-2 in causing myocardial inflammation, which could be alternatively linked to the systemic inflammation state (cytokine storm) and severe hypoxia with cardiovascular injury [21]. We reported a rare case of histologically proved fulminant myocarditis associated with evidence of SARS-CoV-2 within the myocardium and without severe lung involvement at autopsy.

## Figures and Tables

**Figure 1 biomedicines-11-02101-f001:**
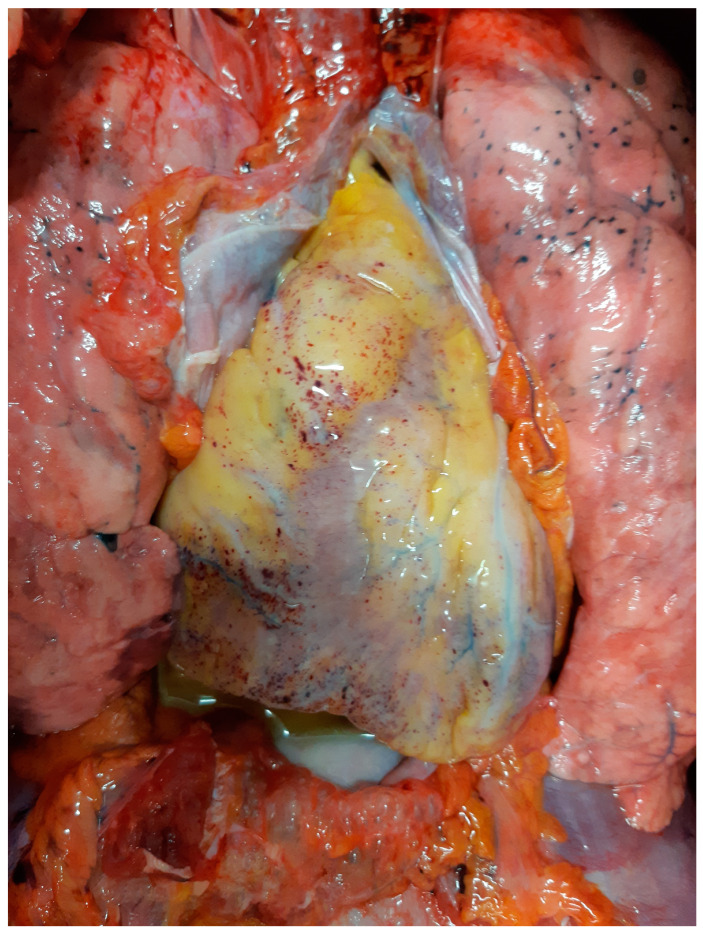
Autopsy findings. Multiple petechiae in the anterior aspect of the heart.

**Figure 2 biomedicines-11-02101-f002:**
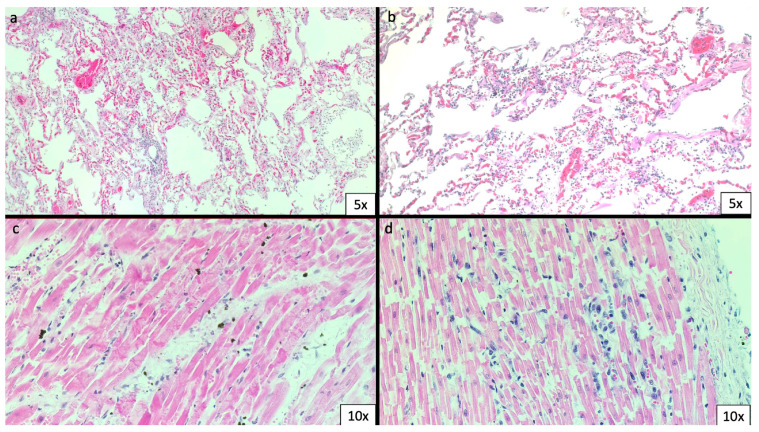
Haematoxylin & eosin (H&E) staining of lungs and heart. (**a**,**b**) Mild leucocytic interstitial infiltration of the lungs. (**c**,**d**) The heart shows areas of contraction band necrosis, erythrocyte spilling and leucocytic infiltration.

**Figure 3 biomedicines-11-02101-f003:**
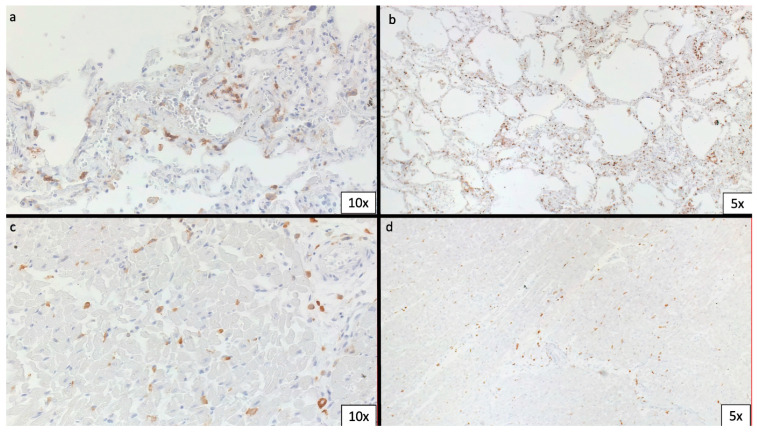
Immunohistochemical staining for CD68 and CD4 in both lungs and heart. (**a**,**c**) Diffuse positivity for CD68 in lung and heart, respectively. (**b**,**d**) Positivity for CD4 in lung and heart, respectively.

**Figure 4 biomedicines-11-02101-f004:**
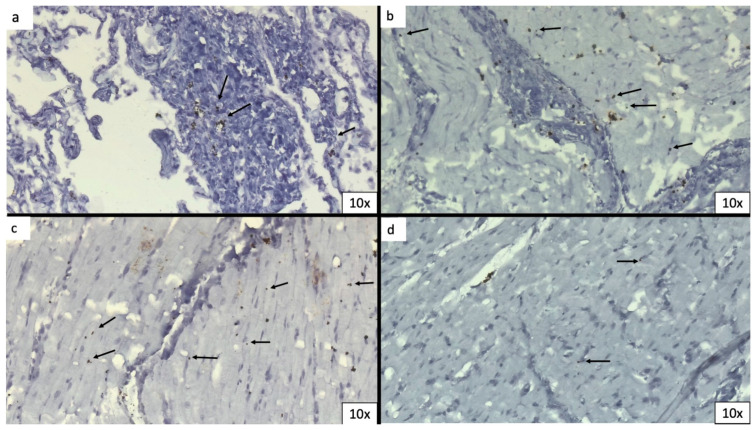
In situ hybridization. In situ hybridization shows viral particles (indicated by the arrow) within the lungs (**a**) and the myocardium (**b**–**d**).

## Data Availability

All data are in the availability of the authors.

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
