# Peer review of "COVID-19-Related Myocarditis: Are We There Yet? A Case Report of COVID-19-Related Fulminant Myocarditis"

_biomedicines, 2023, doi:10.3390/biomedicines11082101_

Round 1

Reviewer 1 Report

The article entitled “COVID 19 related Myocarditis: are we there yet? A case report of
COVID-19-related fulminant myocarditis” by Pierri A. et al. reported a unique case and the post mortem examination of a young patient died of heart failure in a COVID 19 context.

The main originality is the demonstration of SARS-CoV-2 in the myocardia tissue by immunological techniques. The authors observed a massive pulmonary oedema with lymphocyte infiltration and detected SARS-CoV-2 in myocardial cells. The origin of heart failure is not clearly explained since coronary vessels and valves were reported to be normal. Since it is a unique case, it would be interesting to have positive and negative controls of the immunological tests.

The way the virus reaches the heart is not evoked, is it mediated by leukocyte transportation ? The cause of the cardiac failure may be discussed to a larger extent. Are the myocardial cells the target of the virus or altered by cytokines, reactive oxygen species, complement system activation…..

The English language is comprehensible and grammatically correct.

Author Response

Dear Editors,

We thank the reviewers for their comments on the manuscript. We considered each comment with care and we edited the manuscript accordingly. Thank you for considering our case report for publication. Reviewer’s comments: in italics.  Our Reply: normal characters.

Changes introduced in the revised manuscript: underlined.

The article entitled “COVID 19 related Myocarditis: are we there yet? A case report of
COVID-19-related fulminant myocarditis” by Pierri A. et al. reported a unique case and the post mortem examination of a young patient died of heart failure in a COVID 19 context.

  1. The main originality is the demonstration of SARS-CoV-2 in the myocardial tissue by immunological techniques. The authors observed a massive pulmonary oedema with lymphocyte infiltration and detected SARS-CoV-2 in myocardial cells. The origin of heart failure is not clearly explained since coronary vessels and valves were reported to be normal. Since it is a unique case, it would be interesting to have positive and negative controls of the immunological tests.

AUTHOR RESPONSE: The origin of this rapidly progressive heart failure was related to fulminant myocarditis, which is a disease characterized by inflammation of the myocardial tissue, usually involving the myocytes, interstitium and vascular elements. For this reason, the autoptic findings of normal epicardial coronary arteries and uninjured cardiac valves made the diagnosis of myocarditis more likely. We have added this concept to the discussion. Unfortunately, we do not dispose of positive and negative controls of the immunological tests.

  1. The way the virus reaches the heart is not evoked, is it mediated by leukocyte transportation? The cause of the cardiac failure may be discussed to a larger extent. Are the myocardial cells the target of the virus or altered by cytokines, reactive oxygen species, complement system activation…

AUTHOR RESPONSE: Unfortunately, there are no clear data yet on how the virus could reach the heart. However, various hypotheses have been formulated. For example, as reported in the Discussion, Tavazzi et al. propose the possibility of a viraemic phase or, alternatively, the migration of infected alveolar macrophages in extra-pulmonary tissues, including heart. Here, we reported one of the first case of direct cardiac invasion from SARS-CoV-2, thus supposing that this virus may directly infect and damage myocardial cells. However, as described in the Introduction, various mechanisms of indirect cardiovascular injury have been also postulated.

Reviewer 2 Report

Inasmuch as this issue is highly topical, being no chimed pathological finding was related to SARS-CoV2 mycaarditis - this still deserves attention. 

But, for one, abstract is too short and keywords not listed alphabeticaly. Neither are they MeSH terms.

"Blood biomarkers" from the ln 32 - should be discussed in lenght - it is not cleaar which ones - do you mean biomarkers as in clinical cardiology. Do you reffer to antemortem values?

Fig 1a is completely irrelevant, Figs 1 b and 1c are of poorquality, and I  do not think about the pixel quality - focus should be on those petechiae. 

In the caption to fig 2. magnification data are missing and hematoxylin %& eosin is commonly abbreviated. Also, I'm not sure whether I'm confortable with the adjectiveve "standard".

The same is with fig. 3; no magnification or dye data whatsoever.

Reg. Fig. 4 - there are 4 images (a to d), none of which has been  secluded 

Author Response

Dear Editors,

We thank the reviewers for their comments on the manuscript. We considered each comment with care and we edited the manuscript accordingly. Thank you for considering our case report for publication.

Reviewer’s comments: in italics. Our Reply: normal characters. Changes introduced in the revised manuscript: underlined.

Inasmuch as this issue is highly topical, being no chimed pathological finding was related to SARS-CoV2 mycaarditis - this still deserves attention.

  1. But, for one, abstract is too short and keywords not listed alphabeticaly. Neither are they MeSH

AUTHOR RESPONSE: Abstract has been implemented and keywords are now listed alphabetically. The keyword “autopsy” has been changed to MeSH term “post-mortem examination”.

  1. "Blood biomarkers" from the ln 32 - should be discussed in lenght - it is not clear which ones - do you mean biomarkers as in clinical cardiology. Do you reffer to antemortem values?

AUTHOR RESPONSE: We fundamentally referred to cardiac troponins and natriuretic peptides.

  1. Fig 1a is completely irrelevant, Figs 1 b and 1c are of poor quality, and I do not think about the pixel quality - focus should be on those petechiae.

AUTHOR RESPONSE: figure 1 was modified as requested.

  1. In the caption to fig 2. magnification data are missing and hematoxylin %& eosin is commonly abbreviated. Also, I'm not sure whether I'm confortable with the adjective "standard".

AUTHOR RESPONSE: the abbreviation H&E has been added and the adjective “standard” was removed. Magnification data are added in fig. 2

  1. The same is with fig. 3; no magnification or dye data whatsoever.

AUTHOR RESPONSE: magnification are added in fig 3 too.

  1. Fig. 4 - there are 4 images (a to d), none of which has been secluded

AUTHOR RESPONSE: Fig. 4 has been modified

Round 2

Reviewer 2 Report

I believe this paper is ok now